# Physical Examination Tool for Swollen and Tender Lower Limb Joints in Juvenile Idiopathic Arthritis: A Pilot Diagnostic Accuracy Study

**DOI:** 10.3390/ijerph19084517

**Published:** 2022-04-08

**Authors:** Antoni Fellas, Davinder Singh-Grewal, Jeffrey Chaitow, Denise Warner, Ella Onikul, Derek Santos, Matthew Clapham, Andrea Coda

**Affiliations:** 1School of Health Sciences, College of Health, Medicine and Wellbeing, University of Newcastle, Newcastle 2308, Australia; andrea.coda@newcastle.edu.au; 2Sydney Children’s Hospital Westmead, Sydney 2145, Australia; davinder.singhgrewal@health.nsw.gov.au (D.S.-G.); jeffrey.chaitow@health.nsw.gov.au (J.C.); denise.warner@health.nsw.gov.au (D.W.); ella.onikul@health.nsw.gov.au (E.O.); 3School of Health Sciences, Queen Margaret University, Edinburgh EH21 6UU, UK; dsantos@qmu.ac.uk; 4The Hunter Medical Research Institute, Newcastle 2305, Australia; matthew.clapham@hmri.org.au

**Keywords:** arthritis, juvenile, JIA, clinical examination, physical examination tool, magnetic resonance imaging, lower extremity, foot and ankle

## Abstract

Background: Juvenile idiopathic arthritis (JIA) is the most common rheumatic disease in children, with lower limb involvement highly prevalent. Recent evidence has highlighted the lack of specific lower limb physical examination (PE) tools for clinicians assisting the paediatric rheumatology team in identifying lower extremity disease in patients with JIA. Early clinical detection may lead to more prompt and targeted interventions to reduce lower limb problems in children with JIA. The aim of this pilot study is to provide preliminary data on the diagnostic accuracy of a lower limb PE tool in JIA. Methods: Children with JIA requiring magnetic resonance imaging (MRI) on their lower limb joints per their usual care were eligible. Lower limb joint counts were conducted clinically by a podiatrist and paediatric rheumatologist using the proposed twenty joint per side, PE tool. The PE were compared to MRI assessments completed by two independent paediatric radiologists. Data were analysed using agreement (observed, positive and negative) and Cohen’s kappa with 95% CIs. Results: Fifteen participants were recruited into the study in which 600 lower limb joints were clinically examined. Statistical analysis showed excellent inter-rater reliability between podiatrist and paediatric rheumatologist for both joint swelling and tenderness. Results of the intra-rater reliability of the podiatrist using the PE tool indicated excellent percentage agreements (98.5–100%) and substantial kappa coefficients (0.93–1). The inter-rater reliability between radiological assessments contrasted the PE results, showing low agreement and poor reliability. Comparisons between PE and MRI resulted in poor kappa coefficients and low agreement percentages. The most agreeable joint between MRI and PE was the ankle joint, while the worst performing joint was the sub-talar joint. Conclusion: Results indicate potential clinical reliability; however, the validity and diagnostic accuracy of the proposed PE tool remains unclear due to low kappa coefficients and inconsistent agreements between PE and MRI results. Further research will be required before the tool may be used in a clinical setting.

## 1. Introduction

Juvenile idiopathic arthritis (JIA) is the most common rheumatic condition affecting children and adolescence [1]. Common symptoms after disease onset include joint swelling, tenderness and stiffness [1]. Early clinical detection of active joint disease is of paramount importance, as prolonged disease may lead to further physical problems, such as permanent joint damage, muscle atrophy or weakness and flexion contractures [2]. Physical problems have also shown to reduce the likelihood of children with JIA engaging in regular physical activity compared to healthy counterparts, further impacting their quality of life [3]. The lower limb is affected in at least 50% of patients with JIA, with the hip, knee and ankle the most involved [4]. Foot disease is also a frequent clinical manifestation of JIA, including joints such as the sub-talar, talo-navicular, calcaneo-cuboid and metatarsophalangeal joints [4,5,6,7]. Persistent foot disease may cause structural deformities of the rear and midfoot, leading to excessive pronation and exacerbation of associated foot and ankle problems such as Achilles tendinopathies and plantar fasciitis [6,8]. Clinical detection of active joint disease in the feet is typically more difficult than the knee or ankle, often requiring further investigation such as ultrasound or magnetic resonance imaging (MRI) [9,10]. Moreover, children and adolescents with foot disease may experience a less predictable course of disease and prognosis for disease remission [11]. Thus, careful physical examination (PE) of these lower limb joints, particularly of the feet, plays an important role to detect active disease, prompt early evidence-based interventions or sensitive medical imaging such as MRI, and reduce physical impairment.

PE is currently the gold standard in paediatric rheumatology for assessing joint disease in JIA [12,13]. The British Society of Paediatric and Adolescent Rheumatology management guideline for JIA, recommends a multidisciplinary approach which recognises allied health professionals (AHPs) such physiotherapists and podiatrists as core members [12]. Podiatrists may assist the paediatric rheumatologist (PR) to screen the lower limb, particularly the foot and ankle region. Therefore, a standardised and cost-effective lower limb PE tool that includes lower limb joints commonly affected by JIA may be useful for AHPs and PRs to conduct a reliable screening. A PE tool could be used in conjunction with a PR joint assessment to further optimise the detection of active disease and structural deformities of the feet and ankles. The PE tool may also be useful for early clinical management. Despite the frequently diagnosed foot active joint disease in children with JIA, a recent scoping review outlined that there are no PE tools currently available that comprehensively assess foot joints, such as the sub-talar, talo-navicular, calcaneo-cuboid, metatarsophalangeal, distal and interphalangeal joints [14]. 

### 1.1. Objective

This study aims to obtain preliminary data to investigate the diagnostic accuracy (validity and reliability) of a PE tool for the detection of active lower limb joints in JIA.

### 1.2. Hypotheses

-A lower limb PE tool will display moderate correlation (validity) with MRI in the detection of active joint disease;-Podiatrist and PR clinical examination of lower limb joints will display moderate to high percentage agreements (inter-rater reliability);-Independent radiologist examinations of lower limb MRI scans will display moderate to high percentage agreements (inter-rater reliability);-Podiatrist intra-observer PE of lower limb joints will display high percentage agreements (intra-rater reliability).

## 2. Methods

### 2.1. Study Design

This study has two design components. The first component was to design and test the validity of the PE tool by comparing the PE of lower limb joints to MRI. The second component tested the inter (PR versus podiatrist)- and intra (podiatrist)-rater reliability, and inter-rater reliability of radiologist assessments. Ethics was approved by the Hunter New England Human Research Ethics Committee (16/09/21/4.03). 

### 2.2. Participants

Participants were prospectively recruited using a convenience sampling approach from the Sydney Children’s Hospitals at Westmead and Randwick, and the John Hunter Children’s Hospital between February 2018 to August 2019. PRs DSG and JC previously identified potentially eligible participants through the hospital outpatients’ clinics listed. ***Inclusion Criteria*:** Diagnosed with JIA according to ILAR criteria; aged 5 to 18 years old; needs to have received or is going to receive an MRI scan of their feet/ankles as deemed clinically by their PR. The knee and hip were also included. ***Exclusion Criteria:*** Concomitant musculoskeletal disease, central or peripheral nerve disease and endocrine disorders, including diabetes mellitus; recent trauma not related to JIA; children who experience significant disease flare between time of MRI and PE*;* significant changes to pharmaceutical intervention between time of MRI and PE.

### 2.3. Diagnostic Tools

#### 2.3.1. Physical Examination Tool

The proposed PE tool was adopted from Helliwell et al., (2007) [15]. Their PE tool was suggested for its use in adult patients with rheumatoid arthritis and includes important foot joints such as the sub-talar joint, talo-navicular and calcaneo-cuboid joints that are otherwise excluded from current JIA PE tools [14]. This tool can be easily adapted to a clinical setting and has also been used in podiatry-based research in adult rheumatology [16]. However, the PE tool suggested by Helliwell et al. (2007) focused only on foot and ankle joints, and therefore did not include the hip and knee. The tool also combined distal and proximal interphalangeal joints. Therefore, the proposed PE tool was adopted and modified to include the most important and common lower limb joints affected in children and adolescents with JIA. These include the hip; knee; ankle; sub-talar (rearfoot); talo-navicular and calcaneo-cuboid (midfoot); metatarsophalangeal (forefoot); proximal and distal interphalangeal joints (digits) (Figure 1). The hip is a deep lower limb joint and is therefore more difficult to clinically detect the presence of swelling. For this reason, a pathological hip joint was determined if there was a presence of pain and limited joint range of motion. The PE tool includes the count of twenty joints for each side of the lower limb, giving a total of forty joints. The PE tool is designed as a dichotomous scoring system. The clinicians recorded swelling and tenderness by marking the corresponding joint box, and absent by leaving the joint box empty. The proposed PE tools are shown in Figure 1 (swelling) and Figure 2 (tenderness).

#### 2.3.2. Physical Examination

PE of lower limb joints was completed by PRs (DSG or JC), and podiatrist AF. PE was conducted during a scheduled consult at any of the three outpatient clinics. One PE was carried out by either PR (DSG or JC) and two repeat examinations by same podiatrist (AF). Two lower limb assessments from the podiatrist were obtained to test intra-rater reliability. Each PE took no longer than 5 min to complete and were completed at least 15 min apart to help reduce intra-observer bias. The data collection from DSG or JC were compared with the first attempt from AF to test inter-rater reliability. PE from both the PR and podiatrist were conducted during the same consultation but were completed blinded from each other’s assessments. The assessment of joints for this study required clinicians to physically assess lower limb joints for tenderness and swelling. This was achieved by physically palpating the each joint and passively moving joints through their full range of motion. At the same time, clinicians were able to report any signs of swelling, possible differences in joint size to the opposite limb and redness. As soon as the clinician detected the symptomatic joint, the examination on that specific joint was ceased to promptly limit an unnecessary painful experience. The PE results were recorded using the proposed PE tools (Figure 1 and Figure 2). 

MRI was used to test the accuracy of the modified PE tool as it has shown it to be the most sensitive imaging modality to diagnose joint synovitis in JIA [17,18]. It has also shown to detect subclinical disease in JIA, which may have a role in the predicting disease course, flare and joint destruction [19]. As specified within the inclusion criteria, participants were not prescribed an MRI as part of the study. Instead, PRs (DSG and JC) provided details of the study to those patients needing an MRI to further investigate the presence of active disease in lower limb joints. Study information sheets contained contact information of the chief investigator (AF). If consent was given from parents, then a copy of the MRI (either electronic or CD-ROM) scans was acquired for research purposes only. Once consented into the study, a consultation was also arranged to conduct the lower limb assessments. After participants had completed both their MRI scans and the PE, parents were directly contacted if participants had any changes to their medications, joint status or had experienced any joint flares between the time of their MRI scan(s) and PE. The days between PE from the MRI scans were also recorded. Two experienced radiologists in paediatric rheumatology (DW and EO) independently received a copy of the participant MRI scans to assess. As not at all MRI scans were accompanied with a contrast of gadolinium, radiologists were asked to assess joints for effusion, and if contrast was available then an assessment of synovitis was conducted. Therefore, some participant’s MRI scans were assessed for joint effusion only and those with contrast MRIs were assessed for both effusion and synovitis. Joint effusion and synovitis were assessed independent of each other and compared separately to PE results. Both radiologists were blinded from each other and the PE findings. Clinicians conducting a PE were blinded to the results of the MRI. Team members AC and DS were independently responsible for receiving and extrapolating the data from the PE and radiology assessments, respectively. Figure 3 depicts the study flow chart from eligibility through to data analysis. 

### 2.4. Sample Size

This pilot diagnostic accuracy study aimed to recruit approximately 20 participants. This study will be used to power a future larger clinical trial with children diagnosed with JIA. 

### 2.5. Statistical Analysis

All statistical analyses were designed by the study’s biostatistician MC and then programmed using Stata v14.0 (StataCorp Ltd., College Station, TX, USA). Participant characteristics are described with means, standard deviations (SD) or frequencies and percentages. Agreement was assessed with percent agreement (observed, positive and negative) and Cohen’s kappa with 95% CIs. Cohen suggested the kappa result be interpreted as follows: values ≤ 0 as indicating no agreement and 0.01–0.20 as none to slight, 0.21–0.40 as fair, 0.41–0.60 as moderate, 0.61–0.80 as substantial, and 0.81–1.00 as almost perfect agreement. An overall kappa for each comparison was determined by combining all left and right-side joints of participants and treated as independent observations. Indication of disease by diagnosis method and operator is provided with frequencies and percentages.

## 3. Results

A total of 15 participants with JIA receiving an MRI on their lower limb joints were enrolled into the study. The characteristics of these participants are listed in Table 1. The health status of participants recruited in this study did vary including self-reported pain (range 3–18) and quality of life (range 35.22–87.5). Overall, 70% of participants had a diagnosis of polyarticular JIA but a range of medications in this study is noted with at least 30% of participants on combination drug therapies. 

### 3.1. MRI Scans

Overall, 17 MRI scans were obtained and examined in this study. Two participants had bilateral scans of their lower limb joints. Seven out of seventeen MRI scans were contrasted with gadolinium and therefore radiologists DW and EO were able to assess these joints for synovitis as well as effusion. No participants in this study required MRI scans for their hip and/or knee joints. Overall, 195 and 141 joints were scanned and assessed for joint effusion and synovitis, respectively. Table 2 displays the count of present or absent joint effusions and joint synovitis with their corresponding PE results. 

### 3.2. Physical Examination

All 15 participants were clinically examined for joint swelling and tenderness. The total of joints assessed in the PE component for this study was six hundred. The presence and absence of clinical joint swelling and tenderness results are available in Table 2. The sampling technique was convenience and as such the MRI scans and PE were not necessarily conducted on the same day. The average days between MRI and PE was 16.9 days. No participants experienced joint flares or changes to medication between time of MRI and PE. 

### 3.3. Diagnostic Accuracy

The results of agreements between each variable are presented in Table 3. Observed agreement is expressed as a percentage of agreement between each variable. Kappa coefficients and CIs were also produced for each comparison. Table 3 is divided into the agreement between MRI and PE. More specifically: joint effusion versus clinical swelling; joint effusion versus clinical tenderness; joint synovitis versus clinical swelling; and joint synovitis versus clinical tenderness. Table 3 also contains the results of the reliability component to this study specifically the inter-rater agreements between PR and podiatrist for joint swelling and tenderness, the intra-rater agreements between podiatric assessments for joint swelling and tenderness, and the inter-rater agreements between radiologists for joint effusion and synovitis. 

### 3.4. Joint Effusion versus Physical Examination 

Radiologist 1 reported a higher prevalence of joint effusion compared to the PR and podiatrist assessments for clinical swelling and tenderness. The observed agreement between radiologist 1 and PR assessments for clinical swelling and tenderness were both 50.3%. The results were very similar when comparing radiologist 1 joint effusion to podiatrist assessments for clinical swelling and tenderness. In contrast, radiologist 2 had high agreements with both PR and podiatrist when comparing joint effusions on MRI to clinical joint swelling and tenderness. 

### 3.5. Joint Synovitis versus Physical Examination

The comparison between PR joint swelling and radiologist 1 joint synovitis represented the lowest observed agreement (46.8%) and kappa coefficient (0.06) in this study. The podiatrist observed agreement was also low at 50.4%. Clinical joint tenderness versus MRI synovitis agreement remained low between radiologist 1 and/or PR (53.2%)/podiatrist (51.1%). Similar to effusion, radiologist 2 and PR and podiatrist had high observed agreements in both swelling and tenderness versus MRI synovitis. There was 87% observed agreement from both PR and podiatrist when comparing joint tenderness to MRI synovitis. Clinical joint tenderness assessments from both podiatrist and PR, had the highest agreements and kappa coefficients in this study when comparing MRI to clinical assessments. 

### 3.6. Inter-Rater Agreement between Paediatric Rheumatologist and Podiatrist

The inter-rater reliability of PR and podiatrist was tested. Using the same PE lower limb tool, the observed agreement between PR and podiatrist was very high. There was 96.5% observed agreement for joint swelling and 91.5% for joint tenderness. This translated to substantial kappa coefficients of 0.73 and 0.61 for swelling and tenderness, respectively.

### 3.7. Intra-Rater Agreement of Podiatrist

The repeatability of the PE tool was tested by 2 independent clinical assessments by the podiatrist. Observed agreements between attempts 1 and 2 were 100% and 98.5% for joint swelling and tenderness, respectively. Kappa coefficients were almost perfect at 1 and 0.93. 

### 3.8. Inter-Rater Agreement between Radiologists

Radiologist interpretations of joint effusion and synovitis were compared. The observed agreement for joint effusion between radiologists was 50.3 with a kappa of 0.13. The highest observed agreement of joints for joint effusion between radiologists were the ankle joints (80%), and worst joints were the first proximal inter-phalangeal (17%), sub-talar (35%), talo-navicular (38%) and calcaneo-cuboid (44%) joints. The highest observed agreement of joints for joint synovitis between radiologists were the first metatarsophalangeal (67%), ankle (60%) and talo-navicular (60%) joints, and worst joints were the distal and proximal inter-phalangeal (46.5% and 48.8%), sub-talar (50%), and calcaneo-cuboid (50%) joints.

### 3.9. Positive and Negative Agreement

Table 3 lists the positive and negative agreement percentages of each comparison with all joints combined. In every comparison negative agreement produced higher percentages than positive agreement. On average, there was a 38.06% positive agreement where radiologists and physical examiners both agreed in the presence of a pathological joint. In contrast, there was an average 71.69% negative agreement where radiologists and physical examiners both agreed in the absence of a pathological joint. Radiologists and physical examiners were therefore 33.63% more likely to both agree that a joint was absent of disease versus agreeing in the presence of a pathological joint.

Table 4 provides agreement percentages of comparisons for the most commonly involved joints in the study. These include the ankle, sub-talar, calcaneo-cuboid and talo-navicular joints. The phalangeal joints were not included due to the low prevalence in this study. Individual kappa coefficients were also not provided for each joint due to the limited data acquired for each region. The most well performing joint on average across all comparisons was the ankle joint, with an average observed agreement of 67.5% and positive agreement of 72.7%. The calcaneo-cuboid joint had the highest average negative agreement of 62%. The worst performing joint across all agreement percentages was the sub-talar joint, with an average observed agreement of 54.2%, positive agreement of 45.3% and negative agreement of 53.7%.

## 4. Discussion

To our knowledge, this is the first study investigating the accuracy of a lower limb PE tool in children with JIA. Results of this pilot study indicate significant discrepancies between clinical and MRI assessments of foot disease in children with JIA. Radiologist 1 detected higher counts of joint effusion and synovitis compared to a PR or podiatrist. On average, radiologist 1 reported 40% more pathological joints compared to the physical examiners indicating that joints were pathological based on MRI scans but were not detected clinically. However, this was not the case when comparing radiologist 2 with a PR or podiatrist. Radiologist 2 diagnosed a considerably smaller number of pathological joints compared to radiologist 1. There were no instances where radiologist 2 diagnosed presence joint effusion when radiologist 1 reported absence. However, there were four instances where radiologist 2 reported presence of joint synovitis when radiologist 1 reported absent. Statistical analysis confirmed that with the inter-rater reliability between radiologist 1 and 2, producing low percentage observed agreement and kappa coefficients. This may suggest why the observed agreement and kappa coefficients between radiologist 1 and physical examiners were poor but were good between radiologist 2 and physical examiners. 

The discordance between MRI and PE in this study is consistent with previous lower limb studies conducted in JIA [20,21,22]. The similar results have also been reported when comparing US to PE [9,23]. Inter-rater reliability between PR and podiatrist examinations suggests excellent clinical reliability when using the PE tool. An observed agreement of 96.5% and 91.5% for clinical joint swelling and tenderness, respectively, was a promising observation in this study. The same could be said for repeatability. Intra-rater agreement of the podiatrist assessment revealed excellent reliability when repeating clinical joint examinations with the PE tool. Another interesting finding in this study was that every comparison had an equal or higher negative agreement than a positive agreement and, in some cases, there were large disparities. Results were consistent in that radiologists and physical examiners were more likely to agree on a joint that is absent of disease than agreeing on a joint that is pathological. This also includes inter-rater reliability of radiologist examinations. Both radiologists agreed 60% or more of the time that joints did not contain an effusion or synovial enhancement. However, the same cannot be said for positive agreement, with radiologists displaying poor positive agreement for effusion and synovitis. 

When comparing agreement percentages of the most commonly involved joints, the ankle joint was the most consistently agreeable lower limb joint between radiological and physical examinations. It was also the most agreeable joint when comparing the PE from the PR and podiatrist. Despite the sub-talar joint showing some clinical reliability between PR and podiatrist examinations, it did not translate to good agreements between physical examiners and radiologists. On average, the sub-talar joint was the worst performing joint with results displaying low positive and negative agreements between physical examiners and radiologists. The most agreeable joint in the study is also a joint that is easily accessible and examined clinically, while the least agreeable joint was the sub-talar joint, which is considered more difficult to assess, due to its deeper lying position and multiple articulations. 

Inter-rater discrepancies between radiological assessments were an important finding in this study. First, a period of calibration for radiologists prior to the study’s commencement was not utilised in this study, which may have reduced the discrepancy between radiological assessments. Second, there is limited to no evidence of an MRI standardised screening tool for the feet and ankles in JIA. The Juvenile Arthritis MRI Scoring (JAMRIS) system has been developed for the knee and wrist in JIA but not the feet and ankles [24,25]. A JAMRIS system for foot and ankle joints may assist in increasing the inter-rater reliability of radiological assessments. Third, evidence from this study suggests that diagnosing pathological rear and midfoot joints may be more difficult for radiologists to agree on. This is evident with a lack of agreement for sub-talar, talo-navicular and calcaneo-cuboid joints. Lastly, radiologists provided only one attempt in their assessments of joint disease in this study. Miller et al. (2009) showed that any assessment after the first attempt can increase inter-rater reliability and therefore may have improved reliability in this study [18].

Overall, based on discrepancies of agreements between radiologists and clinical examiners, the validity of a PE tool for diagnosing and monitoring pathological lower limb joints in JIA remains unclear. This study has shown that a lower limb PE tool may be reliable for clinical examiners and the same examiner to produce consistent clinical joint assessments; however, this also requires further research. 

### 4.1. Limitations 

Some limitations should be considered while interpreting the results. First, the small sample size would have likely affected the results. We also did not achieve our recruitment target of 20 participants, due to time and funding constraints. Second, as participants required an MRI to be included in the study there is a risk of selection bias. There were no participants included in this study that received an MRI on their hips and/or knees. The lack of MRI referrals for the hip and knee joints may be attributable to selection bias towards foot and ankle referrals. Hip and knee joints may also have required less MRI referrals during recruitment compared to foot and ankle joints due the anatomical complexity of intrinsic foot joints such as the sub-talar and midfoot joints requiring further investigation. Therefore, results only pertain to the feet and ankles and may not be generalisable to the hips and knees. In future studies the hips may require to be analysed separately in a sub-group analysis as the method of clinical diagnosis differs to the remaining lower limb joints. The use of gadolinium enhancement in some MRI scans and not in others may have affected the results of validity of the tool. The PE and MRI scans were not conducted on the same day; therefore, it might have been reasonable to expect changes to the participant’s overall joint health or symptoms. However, this was unlikely to occur in this study, as the average time between MRI and PE was 16.9 days, and no participants had a change of medication or experienced joint flares between the date of PE and MRI. Moreover, one recent study comparing PE of the knee to MRI in children with JIA had a median of 38 days between MRI and PE [26]. A subgroup analysis in the same study found that the time taken between MRI and PE did not affect or improve the diagnostic accuracy [26]. Not all MRIs were conducted with same scanner; therefore, there may have been variability in the quality of images taken. However, eight out of fifteen participants (53%) were scanned at The Children’s Hospital at Westmead, while the remaining seven were scanned in private facilities. The primary reason for differences in location was the waiting list for patients at the hospital to have an MRI scan. Another limitation was the disparity between radiologist assessments. This disparity was attributable to a few key factors which were highlighted and options to limit this disparity elucidated earlier in the discussion section. Regarding intra-observer reliability, there may have been a risk of observer bias given the small sample size and a short recall time. Therefore, readers should evaluate those results with caution. We also did not test the intra-rater reliability of PR assessment of joints, which would be an interesting outcome to evaluate in future studies.

### 4.2. Clinical Implications and Directions for Future Research

The proposed PE tool has shown to be reliable between clinicians conducting a PE on lower limb joints on children with JIA. However, the validity of this tool remains unclear and therefore its clinical application is subject to further research. Given the discrepancy between MRI and PE, it is important clinicians conduct a thorough PE of the lower limb and refer for imaging if clinical assessment is inconclusive. Furthermore, development and validation of a gold standard MRI assessment of foot and ankle joints in JIA is required. Once substantiated, it can then be used on a larger sample size to test the sensitivity and specificity of the PE tool to determine its true validity for the detection and diagnosis of pathological lower limb joints in JIA. 

## 5. Conclusions

The diagnostic accuracy of the tool remains unclear as data produced poor kappa coefficients and inconsistent percentage agreements between MRI and PE. Results of this pilot study indicates promising clinical inter-rater reliability between podiatrist and PR when assessing the lower limb with the proposed PE tool; however, methodological improvements are needed to establish reliability between radiologist assessments. Data from this pilot study will aim to guide future similar studies in improving methodological rigour and validation prior to the tool’s clinical implementation. 

## Figures and Tables

**Figure 1 ijerph-19-04517-f001:**
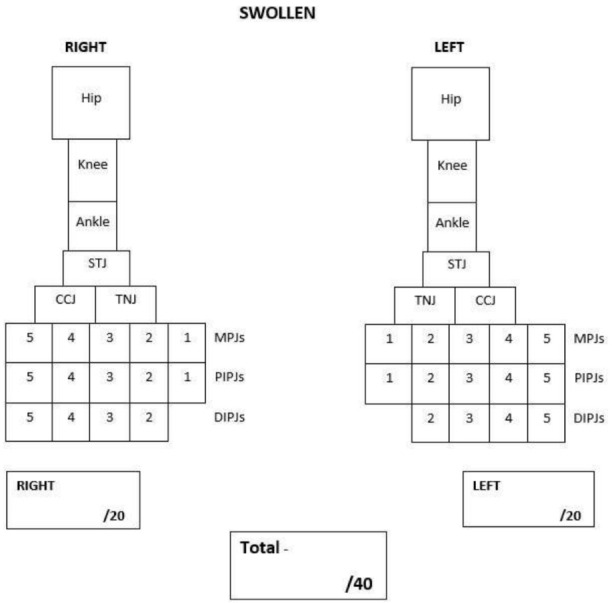
Left and right lower limb swollen joint count.

**Figure 2 ijerph-19-04517-f002:**
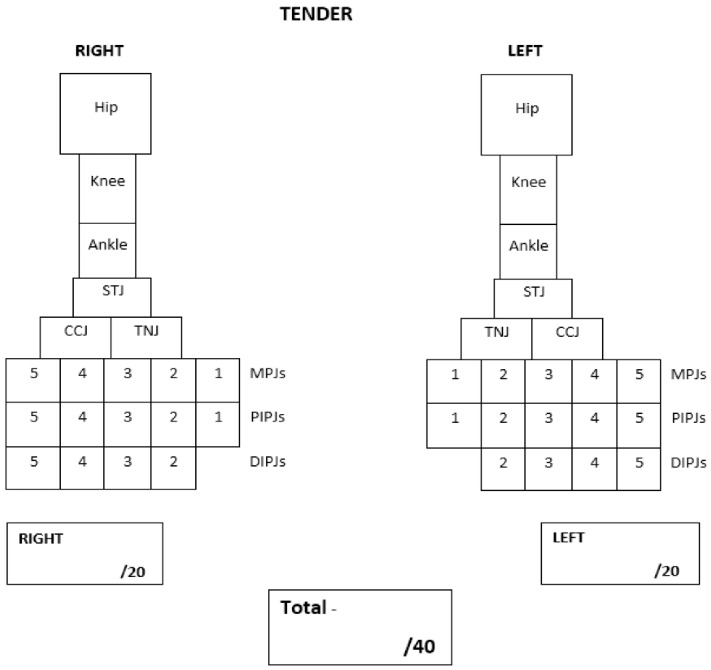
Left and right tender lower limb joint count.

**Figure 3 ijerph-19-04517-f003:**
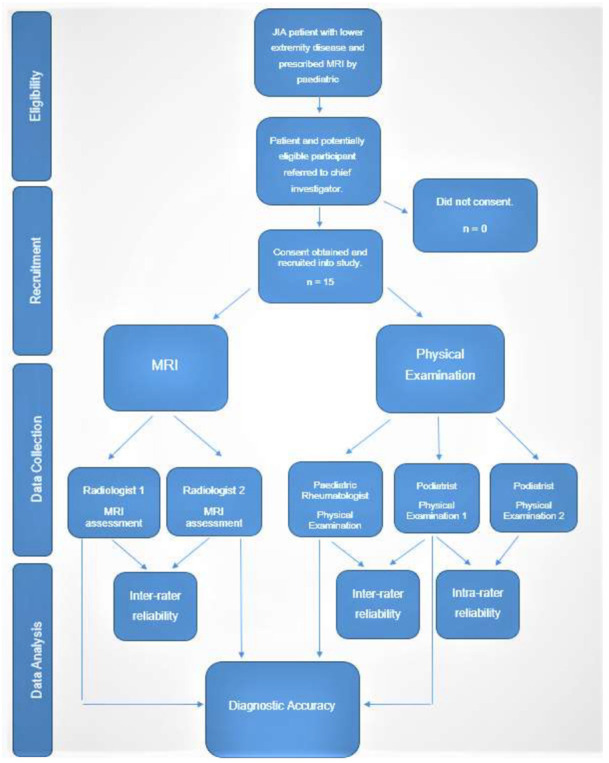
Study Flow Chart.

**Table 1 ijerph-19-04517-t001:** Characteristics of consenting participants.

Characteristic	Participants n = 15
**Demographics**	
Age, years, median (IQR)	11 (6)
Male/female, n	5/10
**Health Status**	
VAS child reported pain, median (range)	48 (3–81)
PedsQL child reported QoL, median (range)	62.50 (35.22–87.5)
Duration of disease, years median (IQR)	3 (3)
**Drug Therapies**	
NSAIDS, n (%)	2 (13)
Methotrexate, n (%)	11 (73)
Etanercept, n (%)	1 (7)
Adalimumab, n (%)	3 (23)
Tofacitinib, n (%)	1 (7)
Prednisone, n (%)	1 (7)
Tocilizumab, n (%)	2 (13)
Leflunomide, n (%)	1 (7)
Combination Therapy—NSAID and Methotrexate and/or Biologic, n (%)	1 (7)
Combination Therapy—DMARD and Biologic, n (%)	3 (23)
**ILAR Subtypes**	
Persistent Oligoarticular, n (%)	1 (7)
Extended Oligoarticular, n (%)	3 (23)
Polyarticular RF −ve, n (%)	7 (47)
Polyarticular RF +ve, n (%)	3 (9)
Psoriatic, n (%)	0 (0)
Systemic, n (%)	1 (7)
Enthesitis-Related, n (%)	3 (23)
Undifferentiated, n (%)	0 (0)

IQR: interquartile range; VAS: visual analogue scale; PedsQL: paediatric quality of life 3.0 rheumatology scale; QoL: quality of life; NSAIDs: non-steroidal anti-inflammatory drugs; DMARD: disease modifying anti-rheumatic drugs; RF: rheumatoid factor.

**Table 2 ijerph-19-04517-t002:** Imaging and clinical results by joint effusion and synovitis, and physical examination.

Imaging and Clinical Findings	Present, n (%)	Absent, n (%)
**MRI—Joint Effusion (n = 195)**		
Radiologist 1	114 (58.46)	81 (41.54)
Radiologist 2	17 (8.72)	178 (91.28)
PR joint swelling	25 (12.82)	170 (87.18)
Pod joint swelling	27 (13.85)	168 (86.15)
PR joint tender	37 (18.97)	158 (81.03)
Pod joint tender	40 (20.51)	155 (79.49)
**MRI—Joint Synovitis (n = 141)**		
Radiologist 1	83 (58.87)	58 (41.13)
Radiologist 2	20 (14.18)	121 (85.82)
PR joint swelling	18 (12.77)	123 (87.23)
Pod joint swelling	21 (14.89)	120 (85.11)
PR joint tender	24 (17.14)	116 (82.86)
Pod joint tender	24 (17.14)	116 (82.86)
**Physical Examination—Joint Swelling (n = 600)**		
PR	41 (6.83)	559 (93.17)
Pod	44 (7.33)	556 (92.67)
**Physical Examination—Joint Tenderness (n = 600)**		
PR	68 (11.33)	532 (88.67)
Pod	81 (13.50)	519 (86.50)

PR: Paediatric Rheumatologist; Pod: Podiatrist.

**Table 3 ijerph-19-04517-t003:** Statistical analysis results for all comparisons.

Comparison	Observed Agreement (%)	Positive Agreement (%)	Negative Agreement (%)	Kappa	95% Confidence Interval
**Joint Effusion (MRI) vs. Swelling (PE)**					
R1 vs. PR	50.3	30	61	0.12	0.04, 0.19
R1 vs. Pod	52.3	34	63	0.15	0.07, 0.23
R2 vs. PR	87	38	48	0.31	0.11, 0.51
R2 vs. Pod	85.6	36	92	0.29	0.09, 0.48
**Joint Effusion (MRI) vs. Tenderness (PE)**					
R1 vs. PR	50.3	26	59	0.10	0.01, 0.19
R1 vs. Pod	51.8	39	60	0.12	0.03, 0.22
R2 vs. PR	79	26	88	0.16	−0.002, 0.32
R2 vs. Pod	80.0	32	88	0.22	0.06, 0.38
**Joint Synovitis (MRI) vs. Swelling (PE)**					
R1 vs. PR	46.8	26	37	0.06	−0.03, 0.15
R1 vs. Pod	50.4	33	61	0.12	0.02, 0.21
R2 vs. PR	84	42	91	0.11	0.11, 0.55
R2 vs. Pod	86.5	54	92	0.46	0.25, 0.66
**Joint Synovitis (MRI) vs. Tenderness (PE)**					
R1 vs. PR	53.2	39	62	0.16	0.06, 0.26
R1 vs. Pod	51.1	36	61	0.12	0.02, 0.23
R2 vs. PR	87	59	92	0.52	0.32, 0.71
R2 vs. Pod	87.1	59	92	0.52	0.32, 0.71
**Physical Examination Pod vs. PR**					
Swelling	96.5	75	98	0.73	0.63, 0.84
Tenderness	91.5	66	95	0.61	0.51, 0.71
**Physical Examination Pod Intra-rater**					
Swelling	100	100	100	1	
Tenderness	98.5	95	99	0.93	0.89, 0.98
**MRI Examination R1 vs. R2**					
Effusion	50.3	26	63	0.13	0.07, 0.19
Synovitis	49.7	31	60	0.11	0.01, 0.20

MRI: magnetic resonance imaging; PE: physical examination; R1: radiologist 1; R2: radiologist 2; PR: paediatric rheumatology; Pod: Podiatrist.

**Table 4 ijerph-19-04517-t004:** Observed, positive and negative agreements of most prevalent joints.

Comparison	Joints											
	Ankle			STJ			CCJ			TNJ		
	OA (%)	PA (%)	NA (%)	OA (%)	PA (%)	NA (%)	OA (%)	PA (%)	NA (%)	OA (%)	PA (%)	NA (%)
**Joint Effusion (MRI) vs. Swelling (PE)**												
R1 vs. R2	80	82.4	76.9	35.7	18.2	47.1	43.8	40	47.1	37.5	28.6	44.4
R1 vs. Pod	73.3	77.8	66.7	64.2	70.6	54.5	37.5	37.5	37.5	68.8	73.7	61.5
R1 vs. PR	60	70	40	57.1	62.5	50	56.3	58.8	53.3	50	50	50
R2 vs. Pod	80	80	80	42.9	0	60	68.8	28.6	80	56.3	22.2	69.6
R2 vs. PR	66.7	70.6	61.5	50	0	66.7	62.5	25	75	75	33.3	84.6
**Joint Synovitis (MRI) vs. Swelling (PE)**												
R1 vs. R2	60	66.7	50	50	54.5	44.4	50	54.5	44.4	60	71.4	33.3
R1 vs. Pod	50	61.2	28.6	30	46.2	0	60	66.7	50	60	71.4	33.3
R1 vs. PR	70	80	40	20	33.3	0	40	40	40	40	50	25
R2 vs. Pod	70	72.7	66.7	60	50	66.7	50	28.6	61.5	60	60	60
R2 vs. PR	50	61.5	28.6	70	57.1	76.9	70	40	80	40	25	50
**Swelling (PE)**												
Pod vs. PR	86.7	84.6	88.2	93.3	87.5	95.5	76.7	46.2	85.1	86.7	75	90.9
**Tenderness (PE)**												
Pod vs. PR	63.3	64.5	62.1	76.7	63.2	82.9	86.7	81.2	89.5	86.7	83.3	88.9

OA: observed agreement; PA: positive agreement; NA: negative agreement; STJ: subtalar joint; CCJ: calcaneo-cuboid joint; TNJ: talonavicular joint; R1: radiologist 1; R2: radiologist 2; PR: paediatric rheumatologist; Pod: podiatrist; MRI: magnetic resonance imaging; PE: physical examination.

## Data Availability

The datasets used and/or analysed during the current study are available from the corresponding author on reasonable request.

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
