# Peer review of "Physical Examination Tool for Swollen and Tender Lower Limb Joints in Juvenile Idiopathic Arthritis: A Pilot Diagnostic Accuracy Study"

_ijerph, 2022, doi:10.3390/ijerph19084517_

Round 1

Reviewer 1 Report

The Authors have presented sufficient data. The  appropriate tables and figures have been provided. The article is easy to read and logically structured.  The Authors used appropriate statistic methods. The conclusions are consistent with presented evidence and arguments. References are up to date and complete. However, some improvements could be made in order to achieve a better paper

  • The keyword “Juvenil Idiopathic Arthritis" is not a Mesh term, Could you replace to "Arthritis, Juvenile"

  • Introduction seems enough but could achieve a deeper knowledge about state of art.
  • The research hypothesis are not described

  • The total sample is not homogeneous respect gender, could you add something about it ?
  • The complete chronology is missing. Please, complete

Reviewer 2 Report

The aim of this pilot study is to provide preliminary data on the diagnostic accuracy of a lower limb physical examination tool in JIA. 
The study is very well written. Results indicate potential clinical reliability; however, the validity and diagnostic accuracy of the proposed PE tool remains unclear due to low kappa coefficients and inconsistent agreements between PE and MRI results. Of course Further research will be required before the tool may be used in a clinical setting.
I have no comments and Suggestions.

Author Response

Thank you for your kinds words.

Reviewer 3 Report

Thank you very much for the opportunity to review this interesting paper!

To my opinion, providing a physical examination tool/guideline to diagnose and describe foot problems in JAI patients is a very welcomed idea! Of course, it would have been more sound if more - and maybe more independent - observers would have been involved. But given, that it is a first approach to the topic I think it is worth being published.

Thank you.

Author Response

Thank you for your kind words. 

Reviewer 4 Report

The manuscript reports the results of a study to validate a clinical tool to obtain useful, valid, and accurate information in patients with JIA.  The study is relevant because it is compared to magnetic resonance imaging, and that, depending on the results in terms of its validity and reliability, sensitivity, and specificity, could be useful in daily clinical practice.

He then made some comments.

Introduction. The whole manuscript follows the same structure in terms of citations, however, in line 41 reference 10 appears. This must be corrected.

The objective of the work must be specified since what is intended is to assess the validity and reliability of the tool. Although it must be specified precisely in the "material and methods" apparatus that the tool values.

Patients and Methods. It is indicated that the intra-rater reliability will be valued, but only with the podiatrist, why is it not interesting to also evaluate this in the PR?  If we understand that the reliability in the use of a physical examination tool can be affected by the experience of the clinician or by their training or previous knowledge, it is very interesting to evaluate the reliability in other professionals, since if the tool is valid and reliable, it is very likely that they will use it. I believe that this is a limitation that should be highlighted.

About the tool, I have not been able to access Figure 1 of the tool, since they do not appear in the manuscript, these must be included to complete the review process completely.  In general, none of the figures appear.

Regarding the assessment of the hip joint, it is suggested in the manuscript (lines 105-108) that the evaluation procedure is different than, in other joints within the tool, modifying the criteria to determine if a joint is pathological, this change in the criteria should be studied separately, that is, determine the validity of this different criterion. In turn, should be highlighted in the limitations.

In lines 118 and 119, about the intra-observer bias, I think it is possible that there is an overvaluation of repeatability because of recall since, in such a small sample, and with only 15 minutes I do not think it can be assured that the intra-observer has been reduced, so it seems to me that the results must be evaluated with caution.  This is extendable to the results section

Because subjects who needed an MRI were included (134-135), there could be selection bias. This would be a limitation to highlight.

In some cases, the effusion was evaluated while in others the synovitis, depending on the use of gadolinium or not in the MRI, which I think may affect the results of validity of the tool.

Although it is a pilot study, I believe that it should be specified how the sample size has been determined (Díaz-Muñoz Gustavo. Methodology of the pilot study. Rev. chil. radiol. 2020; 26( 3 ):100-4.  http://dx.doi.org/10.4067/S0717-93082020000300100. ). Similarly, the type of sampling carried out must be specified.

Results and Discussion.

The degree of agreement between the different radiologists is quite variable, so it would be necessary to determine what influence the experience of these has, or the sub-specialization they may have, or others.  This situation is also reflected in the discussion section, however, I believe that there is a lack of an interpretation of these results, as well as a confrontation with other works, if any.  This situation is repeated on lines 352-357.

Conclusions. As indicated in the manuscript, "the diagnostic accuracy of the tool remains unclear", this should be the main conclusion, ahead of the reliability obtained, since, if the test is not valid, its reliability is relegated to a second place. On the other hand, because there may be an overvaluation of repeatability, this should be clarified in the manuscript.

Reviewer 5 Report

This manuscript entitled “Physical examination tool for swollen and tender lower limb joints in juvenile idiopathic arthritis: a pilot diagnostic accuracy study” aimed to investigate the diagnostic accuracy of a lower limb PE tool in JIA. This paper is well-written and well-structured. However, there are still some issues that should be considered before it can be accepted for publication. Here below are some suggestions.

  1. Abstract, line 14-17, it is suggested that the authors should further highlight the necessity of using PE tools for patients with JIA.
  2. Line 25, “PRs”, abbreviations should be used after it has been explained before.
  3. Please modify and improve the quality of the keywords as this will assist others when they are searching for information on your research topic.
  4. Introduction, Line 40-41, “Common symptoms after disease onset include joint swelling, tenderness and stiffness [10].”, The references should be cited in order.
  5. The authors would like to provide preliminary data on the diagnostic accuracy of a lower limb PE tool in JIA, it is suggested that the “lower limb PE tool” should be further specified in this session, and I notice there are some previous studies that have already assessed the effects of PE tool (18-20, 24), therefore, it is suggested that the authors should further highlight the significance of this study when compared to the previous ones.
  6. It is suggested that the authors could further emphasize the importance of physical activity for children and adolescents and the negative influences of JIA on physical activity, which may help further highlight the significance of this study, here below are some recent studies that could be cited here.
  • Physical activity and sedentary levels in children with juvenile idiopathic arthritis and inflammatory bowel disease. A systematic review and meta-analysis. Pediatric Research, 2019, 86(2), 149-156.
  • Physical Fitness, Dietary Habits and Substance Misuse: A Cross-Sectional Analysis of the Associations in 7,600 Swedish Adolescents. Physical Activity and Health, 2022, 6(1), 26–37.
  • Feasibility and safety of a 6-month exercise program to increase bone and muscle strength in children with juvenile idiopathic arthritis. Pediatric Rheumatology, 2018, 16(1), 1-12.
  • Physical activity and health-related quality of life in children and adolescents: A systematic review and meta-analysis. Health Psychology, 2018, 37(10), 893.
  1. Line 71-72, some hypotheses are suggested to be added at the end of this session.
  2. Methods, line 111-112, “The proposed PE tools are shown in figure 1 (swelling) and figure 2 (tenderness).”, Figures 1 and 2 cannot be found in this manuscript.
  3. Physical examination, it is suggested that a detailed flow chart could help the readers better understand the physical examination process.
  4. Line 156, “This pilot diagnostic accuracy study aimed to recruit approximately 20 participants.”, the sample size would affect the result of statistical analysis. The authors should illustrate whether the sample size of this trial was calculated or just be set according to experience? In addition, the authors aimed to recruit 20 participants, but only 15 were finally included, some explanations should be added.
  5. The results are duplicates on the results session and the tables, and they should be further condensed.
  6. The conclusion should also be further strengthened based on the findings of this study.

Round 2

Reviewer 4 Report

I thank the authors for their efforts.

About the comment “Could it not be done in the same study and presented in an overall lower limb assessment tool? If anything authors could analyse the hip joint in as a sub-group analysis in larger studies.” I agree that it can be done in the same study, but as they indicate it is interesting analyse the hip joint in as a sub-group analysis.

About the comment “The sampling was a convenience sampling which was specified in the “Physical Examination” sub-heading of the results section (line 187)”, indeed, but I consider  that it should appear in the section of “Materials and Methods”.

Author Response

  1. Agreed. Some information was added in the limitations section regarding possibility of a sub-group analysis for hip joints in future studies.
  2. Agreed. Convenience sampling has now been added earlier as well in the methods section. 

Thank you for all your constructive comments. They have really helped strengthen this paper. Your time is much appreciated.

Reviewer 5 Report

Results of this pilot study indicates 513 promising clinical reliability between podiatrist and PR when assessing 514 the lower limb with the proposed PE tool,

Author Response

1. Apologies it wasn't clear whether you wanted that comment amended. A minor word change was added to specify that it was inter-rater reliability that showed promising results. Hope this was ok.